# Damage evolution and constitutive model of the rock masses with non-penetrating cracks under repeated impact loading

Jie Zhang[1]*, Xu Wu[2]

1 China ENFI Engineering Corporation, Beijing, Haidian, China, 2 Beijing Municipal Engineering Research Institute, Beijing, Haidian, China

* zhangjie.01@enfi.com.cn

## Abstract

A large number of cracks exist in natural rock masses, which significantly affects the stability of surrounding rocks in engineering under impact loading. Repeated impact tests by Split Hopkinson Pressure Bar are performed on non-penetrating cracked granite specimens with different prefabricated-crack inclination angles (0, 30, 45, 60, and 90˚). The damage evolution law of cracked rock under repeated impact loading is investigated. Macroscopic damage variables considering geometric and mechanical parameters of cracks are proposed. Further, a constitutive model for the impact loading test is developed based on the coupling damage. It has been found that, the impact resistance of fractured rock first decreases and then increases with the increased prefabricated-crack inclination angle. The impact resistance for specimens with an inclination angle of 45˚ is the minimum. Theoretical results from the developed model agree with the experimental data. The model could well describe the progressive damage characteristics of cracked rock masses.

## 1 Introduction

There is an increasing demand for rock engineering infrastructure constructions, including hydropower, transportation, railway, energy, and mining [1–3]. Natural rock masses contain a large number of discontinuous joints, cracks, and other weak structural interfaces. Moreover, the occurrence, sizes, and distribution of the weak interfaces are normally random, which significantly affects the deformation and failure mechanism of rock masses [4–6]. In engineering practice, fractured rock masses may be subjected to various repeated impact loading such as blasting excavation, mechanical rock drilling, and earthquakes for a long time [7–9]. Most rock masses are subjected to various loads and multiple actions. The repeated impact loading causes cumulative damage to rock masses, which attenuates rock masses and endangers the safety of engineering projects and workers. Therefore, it is necessary to understand the damage evolution law of cracked rock masses under multiple repeated impact loading for the safe construction and evaluation of rock engineering.

The research on the rock masses with non-penetrating cracks shows that the crack deflection and branching phenomenon often occur in the process of crack propagation [10,11]. New

**Funding:** This research was funded by the Beijing Municipal Natural Science Foundation (grant no.

8214049). The project name is Initiation-propagation mechanism and structure safety evaluation of lining cracks under subway vibration load. (Wu Xu's workplace is Beijing Municipal Engineering Research Institute).

**Competing interests:** The authors have declared that no competing interests exist.

cracks in rock specimens under uniaxial compression are initiated at the prefabricated-crack tips [12], and further crack propagation and merging dominate the final failure mode of the rock structure. Antonio [13] studied the effect of prefabricated crack location on crack initiation and propagation and clarified the difference between the wing and secondary cracks. Zhou [14] studied the influence of joint connectivity on the mechanical characteristics of rock masses using the equivalent rock mass (ERM) technology from micromechanics, revealing the connectivity mode of the cracked rock masses. These researches reveal the mechanical response characteristics of fractured rocks under static or quasi-static stress and the law of crack initiation, development, and penetration.

The interconnected cracks significantly affect the dynamic response of surrounding rocks by studying the crack mechanism of cracked rock masses under the blasting stress wave [15]. Zhang [16] studied the crack propagation law of intermittently cracked sandstone specimens under the dynamic and static loads, finding that branch crack propagation and penetration have inertial effects. Wing cracks at the crack tip are easy to directly penetrate at the inner ends of the two prefabricated cracks. Li and Gong [17–19] carried out the dynamic and static loading test of rocks to analyze the change law of dynamic mechanical properties of rocks under different static-load constraints. Li [20]compared the difference in the tensile strength of rock samples under the static and dynamic loads and pointed out that the failure mode and strength of rocks are closely related to the loading rate. Kawamoto [21] applied damage mechanics to the theoretical study of cracked rock masses and described the deformations and crack behaviors of rock masses within the framework of continuum mechanics. Besides, a damage mechanics model of cracked rock masses was established. Ling [22] studied the fatigue damage evolution law of rocks with prefabricated surface cracks under cyclic loading. Liu [23,24] combined the effects of microscopic and macroscopic defects of cracked rock masses on the mechanical properties of rock masses to propose dynamic damage constitutive models for penetrated fractured and non-penetrated fractured rock masses, respectively. Wang [25,26] studied the effect of the confining pressure on the mechanical properties and damage evolution of rock masses under multiple impact loadings to reveal the damage evolution mechanism of rocks containing pre-existing surface cracks.

Although the existing constitutive model of cracked rock masses considers the influences of macro and mesoscopic initial defects on the mechanical properties of rock masses, there are deficiencies in studying the damage evolution during crack propagation under repeated impact loading. The influence of cracks on rock masses needs to consider the macroscopic damage to rock masses caused by the initial crack as well as the rock-mass damage caused by the dynamic propagation of wing cracks. In this paper, granite specimens with non-penetrating cracks in angles 0–90° are prepared. Then repeated impact tests by Split Hopkinson Pressure Bar are carried out. Stress-strain curve characteristics for the specimens are analysed. Further, the damage variables at both macro and meso scales are proposed. A constitutive model coupling the meso and macro damage is developed and verified with experimental data.

## 2 Experimental methodology

### 2.1 Sample preparation

The quasi-static compression test was considered to discuss the damage of rocks' impact loading and their accumulation effect. The same intact granite block was cut and processed into standard cylindrical specimens with a diameter of 50 mm and a height of 100 mm. Then both ends of each sample were smoothed to ensure that their flatness errors were less than 0.02 mm. After the complete rock specimens were processed, semi-elliptical surface cracks with a width of 0.3 mm were made by water jet cutting. The prefabricated cracks were located on both sides

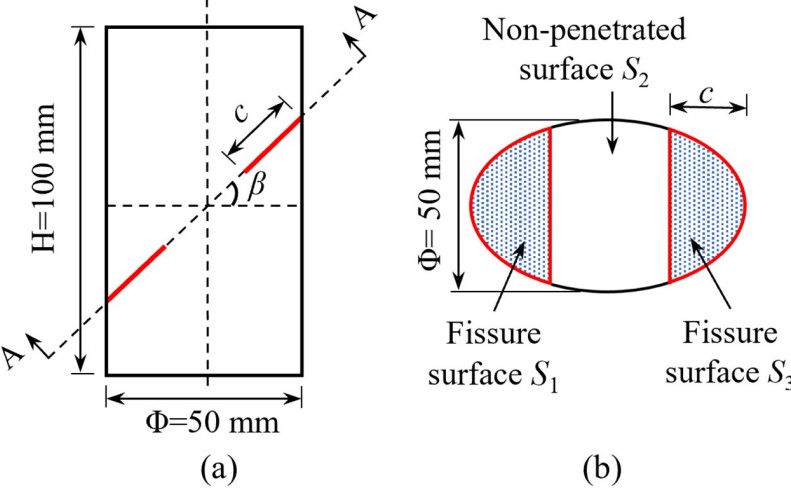

**Fig 1. Schematic diagram of the crack structure of the rock specimens.**

of the specimens, and the front and rear of the cracks were distributed symmetrically in the center. Crack angle $\beta$ was defined as the angle between the loading directions of the specimens and the existing cracks, and the angles were 0, 30, 45, 60, and 90˚, respectively. Three specimens are selected for testing under each working condition (see Fig 1 for the crack structure of the rock specimens).

The connectivity rate is used to characterize the discontinuity of rock masses. Connectivity rate $k$ of a crack surface refers to the ratio of the total area of the crack surface in the cross-section at a certain direction to the area of the entire cross-section. In the test, $k$ is denoted as 0.25; for the area of prefabricated cracks on both sides, $S_1 = S_2$. After calculation of Eq (1), prefabricated crack's length $c$ of rock samples with different inclination angles (0, 30, 45, 60, and 90˚) is 11.5, 13.4, 16.3, 23.0, and 17.5 mm, respectively.

$$\text{Connectivity rate } (k) = \frac{\text{Crack area } (S_1 + S_2)}{\text{Cross-sectional area } (S_1 + S_2 + S_3)} \tag{1}$$

## 2.2 Experimental procedure

The experiment used a separate SHPB device with a bar diameter of 50 mm (see Fig 2 for the separate SHPB device). A compressed nitrogen catapult was used to propel the cone-shaped striker. The length of the incident and the transmission bars was 1,800 mm; that of the buffer bar was 1,000 mm; the diameter of the rod was 50 mm; the length of the cone-shaped striker was 360.1 mm. All were made of high-strength 40Cr alloy steel, with a longitudinal wave velocity of 5,447 m/s, an elastic modulus of 240 GPa, and the uniaxial compressive strength of 800 MPa. The BE120-SAA strain gauges are attached to the incident and transmission bars to collect the incident, reflected, and transmitted wave signals.

In the test of repeated impact loading, rock samples were simultaneously applied with axial static stress $P_0$ and dynamic impact loading $P_d$ with a certain force. Fig 3 shows the stress-wave loading. The single impact load should be within the range of the rocks' dynamic damage threshold to realize the repeated impact of rock specimens. An obvious damage accumulation effect occurred in this range, which could analyze the progressive damage law of rock specimens. Firstly, apply axial pressure to the rock sample at a rate of 0.5 MPa/s until the

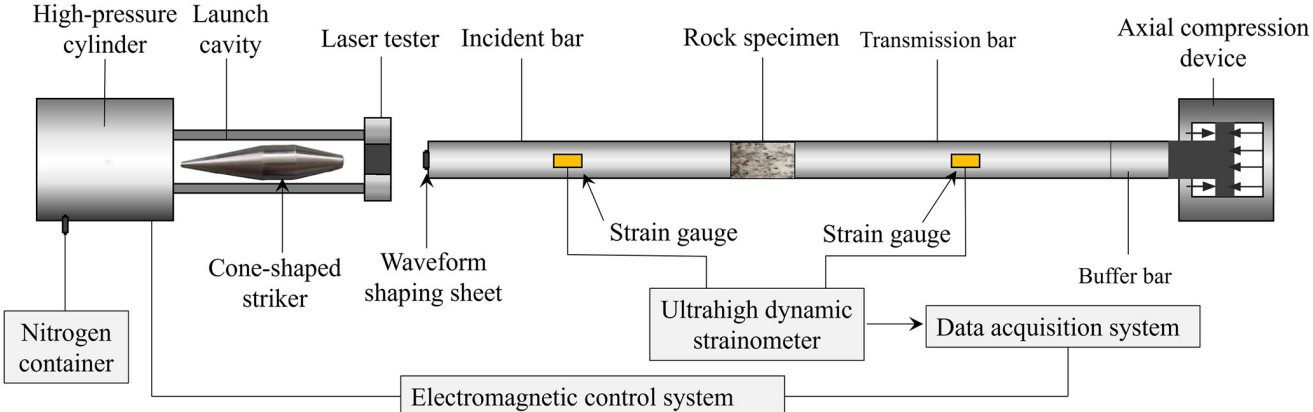

**Fig 2. SHPB device.**

predetermined axial pressure value is reached and remains stable. Then repeatedly impact the rock specimen with the set impact pressure, adjusting the axial pressure value before each impact to ensure consistency with the set value. When the rock is damaged and the test is completed, record and save the data collected after each impact. During the test, axial static stress $P_0 = 8$ MPa, and the impact load needs to be adjusted repeatedly. When the impact air pressure was 0.18 MPa (with an impact velocity of 10.07 m/s), no new cracks appeared in the rock specimens during the first impact. Besides, the cracks expanded and penetrated after several impacts, causing damage.

In the experiment, the incident wave loading lasted approximately 200 μs. The incident, reflected, and transmitted waves were half-sine waveforms. Fig 4 shows the waveforms of granites at the same impact velocity in the SHPB test. The incident waves of multiple impacts have

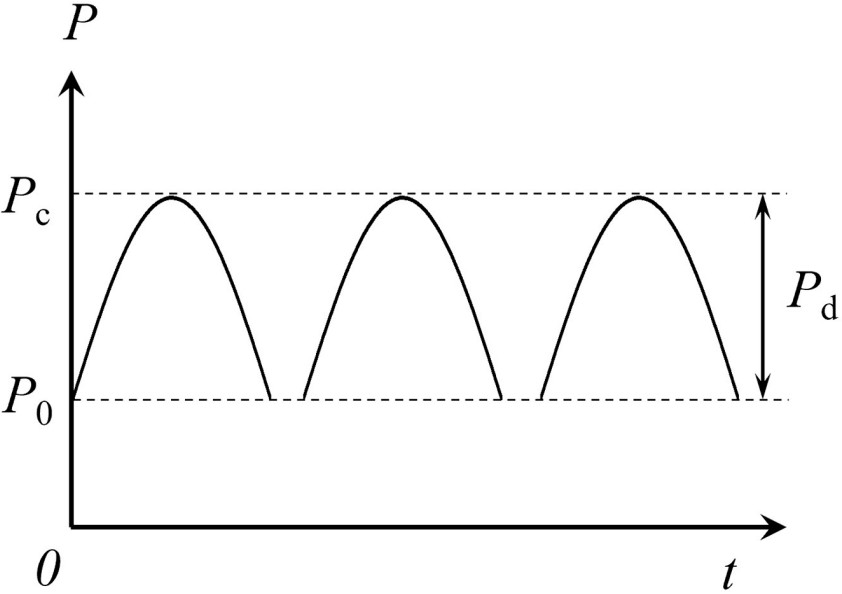

**Fig 3. Stress-wave loading** [27].

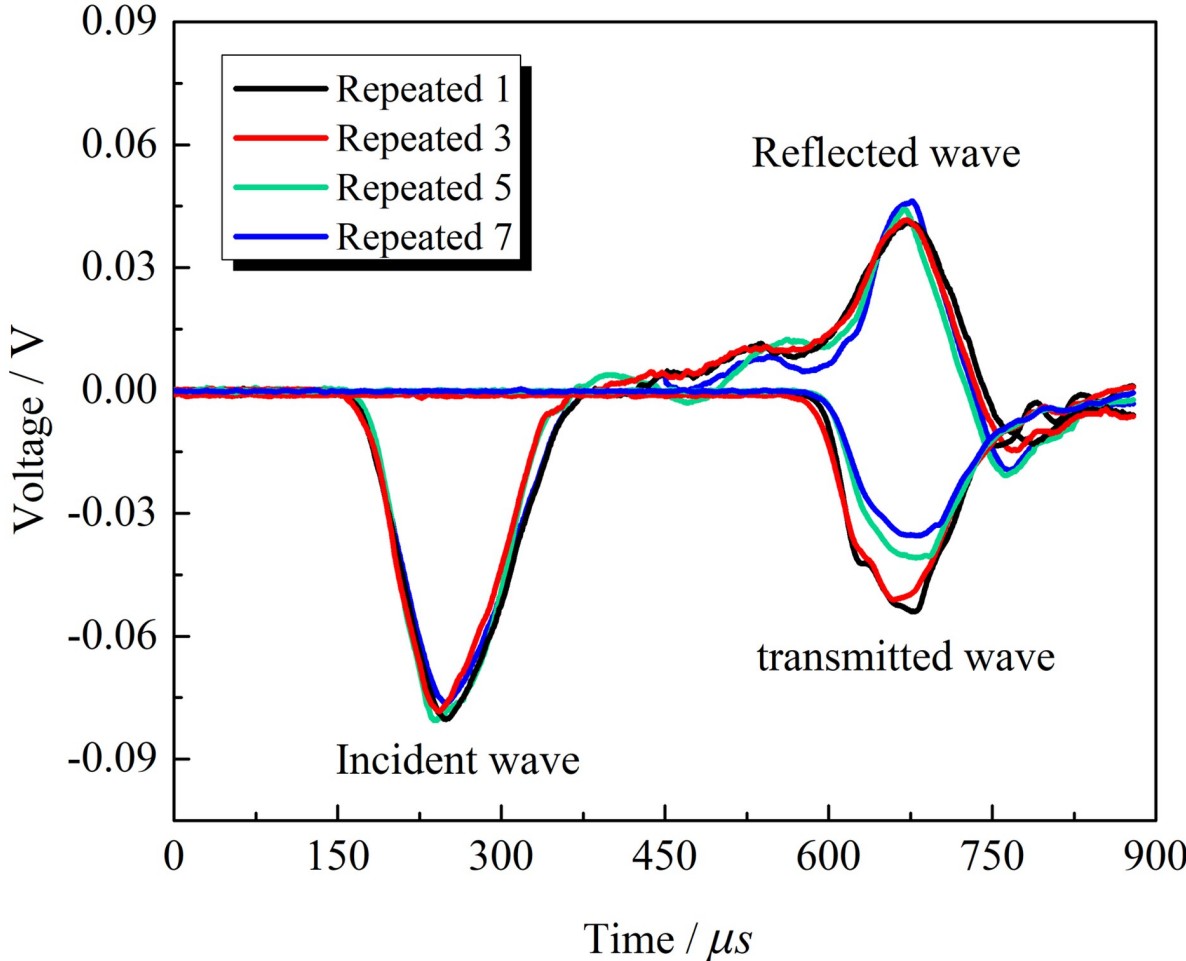

**Fig 4. Waveforms of the repeated impact SHPB test of granite samples.**

good coincidence, indicating the accurate control of the impact velocity. Determine whether the stress balance condition is met based on the degree of overlap between the transmitted wave, incident wave, and reflected wave in the stress balance diagram. Based on this judgment method, it can be seen from Fig 5 that the transmitted wave and superimposed wave basically coincide and remain in the post-peak area for a certain period of time, indicating that the sample can achieve stress equilibrium conditions well during the loading process, proving the effectiveness of this experiment.

The incident wave amplitude at constant impact velocity is the same, while the reflected and transmitted waves had coincident waveforms in the first three impacts, indicating that impacts caused weak damage to rock at the previous stage where micropores and cracks were gestated. The transmitted wave amplitude reached the maximum in the first impact and decreased in subsequent impacts (see Fig 4). The damage degrees of rocks were different at the same impact velocity. Energy consumption of rock's impact damage gradually increased. At the early stage, the rock had a small transmission wave amplitude and little energy consumption. After several impacts, damages accumulated to develop micro-cracks, which increased the energy consumption of damage to rocks.

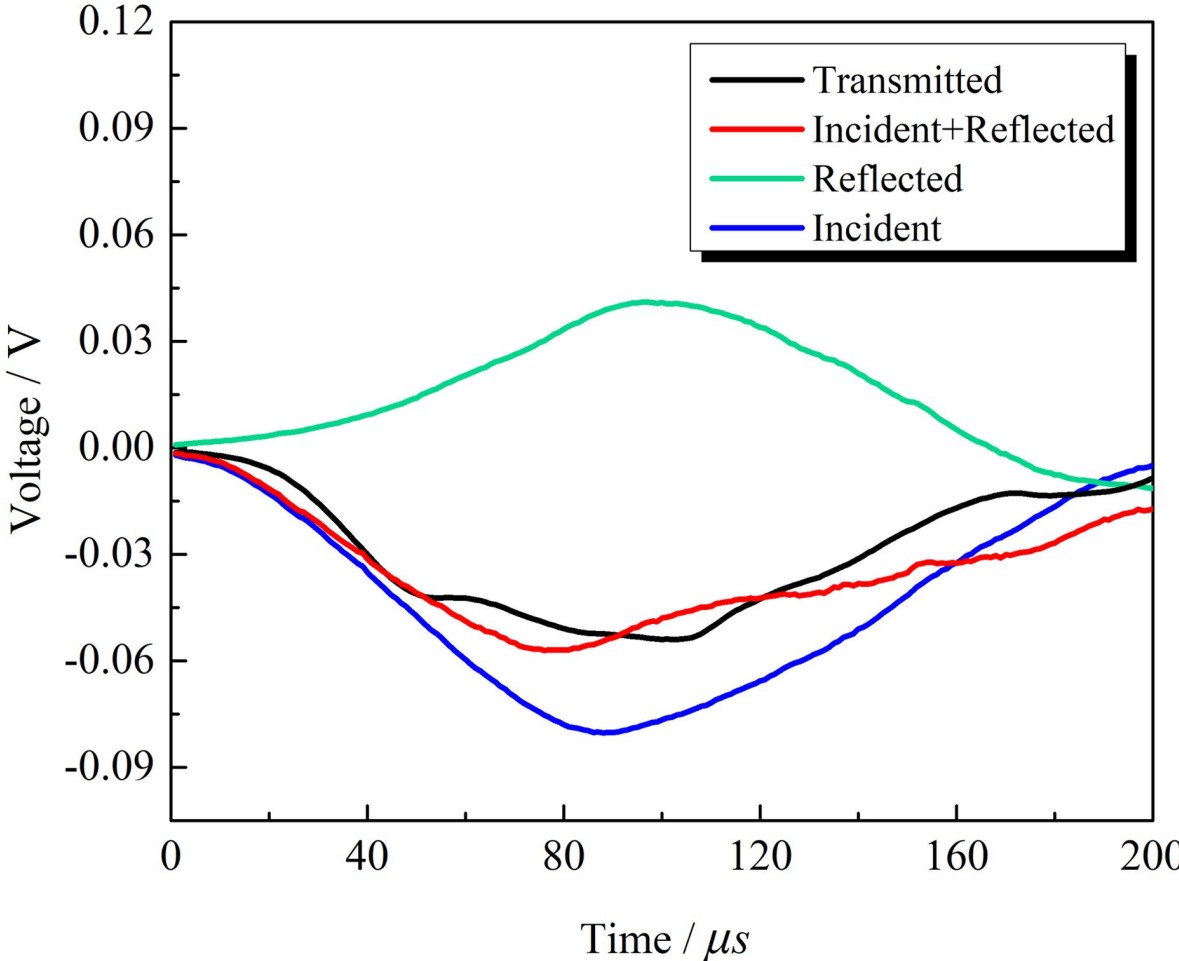

**Fig 5. Stress equilibrium curves of granite samples.**

## 3 Experimental results and analysis

### 3.1 Stress-strain curve characteristics

Fig 6 shows only the stress-strain curves of typical rock specimens with 0˚ and 30˚ cracks' inclination angles under repeated impact loading. The typical dynamic stress-strain curve obtained has no compaction stage. The curves rise non-linearly before the dynamic stress peak and then rebound after the peak. Internal elastic energy of rocks releases at the peak when internal elastic force is greater than unloaded repeated stress. In Fig 6, internal damages accumulate, which degrades the mechanical properties of rock samples with increased impacts. There are differences among the curves of rock samples with different cracks' inclination angles, indicating that the crack structure partly affects the impact resistance of rocks.

The deformation modulus reflects the stress characteristic quantity required for the rock to produce unit strain. The weighted average of the secant modulus, first type secant modulus, and deformation modulus of the loading section is defined as the dynamic deformation modulus, which is used to reflect the compressive deformation characteristics of rocks during the dynamic loading stage. The dynamic deformation modulus of the rock is defined according to the Reference [28] and the relationship between the dynamic deformation modulus and repeated impact

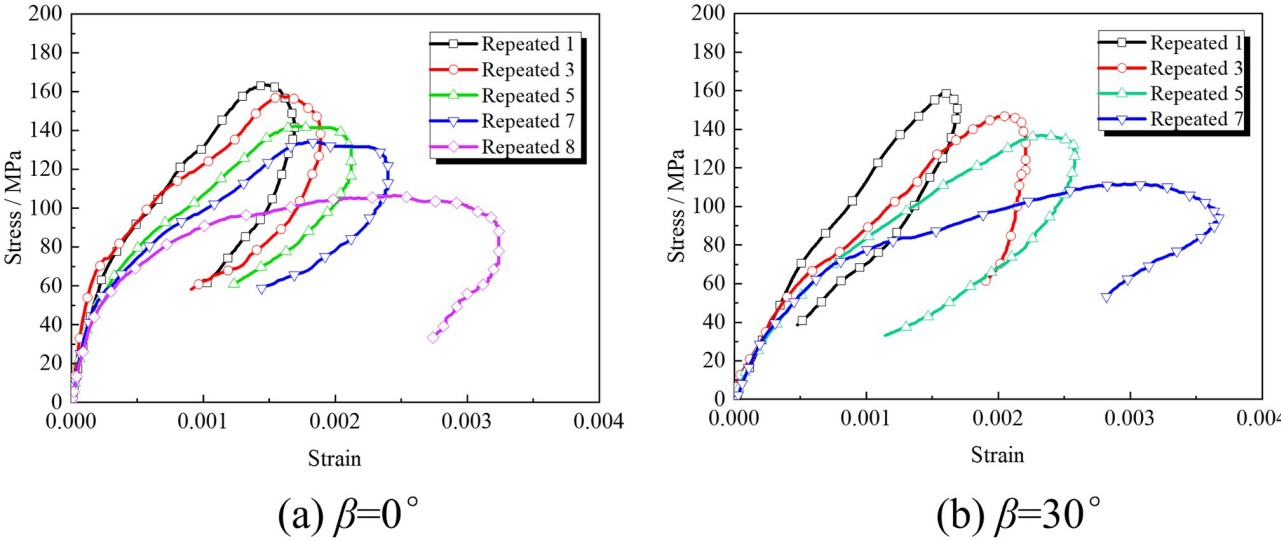

**Fig 6. Stress-strain curves of rock samples with different cracks' inclination angles.**

number at the loading stage based on different cracks' inclination angles are calculated as shown in Fig 7. The test results show that the dynamic deformation modulus decreases with increased impacts because of the cumulative evolution of internal damage during the repeated impact. As impacts increase, the deformability to resist external loads decreases.

The dynamic deformation modulus of granites increases after decreasing with the increased crack's inclination angle (see Fig 7). For rock samples with cracks' inclination angles of 0–30˚, the deformation modulus has a larger reduction amplitude than the previous impacts. The reason may be that rock samples with small inclination angles have small crack lengths. Therefore, prefabricated cracks first close under cyclic-impact loads. When prefabricated cracks with an inclination angle of 0˚ are completely closed, the stress wave can be transmitted without reflection, which inhibits the deterioration of the material [29]. Rock samples with cracks' inclination angle of 45˚ are first damaged after five impacts. The inclination angle and axial loading are more conducive to the development of cracks. Rock samples with an inclination angle of 30˚ have a slightly greater reduction in deformation moduli than that of 60˚. With the strongest impact resistance, rock samples with an inclination angle of 90˚ are least affected by prefabricated cracks.

## 3.2 Damage evolution law of the rock masses with non-penetrating cracks

**3.2.1 Definition of damage variables.** The compositions and combinations of minerals are different In the process of producing rocks, and the strength of the micro-units in the rocks is different. Damage occurs under the internal and external loads, which changes the internal structure and manifests as the germination, expansion, and penetration of microcracks. The damage evolution law of rocks under repeated impact is analyzed according to the damage variables of fractured rock masses of macro and mesoscopic defects. Meso-damage is generalized to be caused by the intact rock in the loading process; the macro-damage is generalized as initial damage caused by the prefabricated macro-cracks to the rocks, and new macro-damage is evolved from the micro-cracks. The sum of the damage strains of these two parts is equal to the strain caused by macro and mesoscopic damage, which conforms to the principle of the equivalent strain calculation in Fig 8 [23].

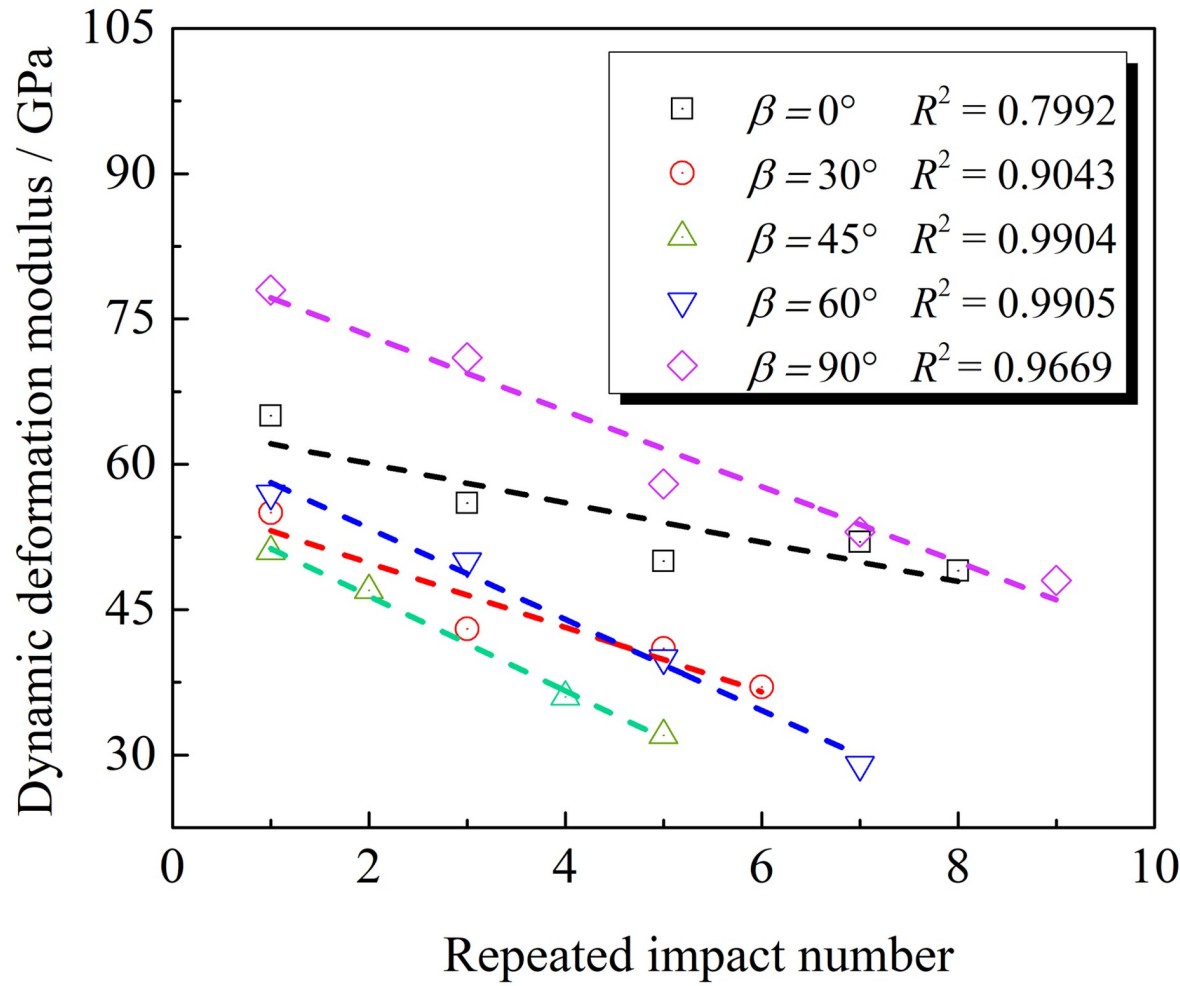

**Fig 7. Relationship between the dynamic deformation modulus and repeated impact number.**

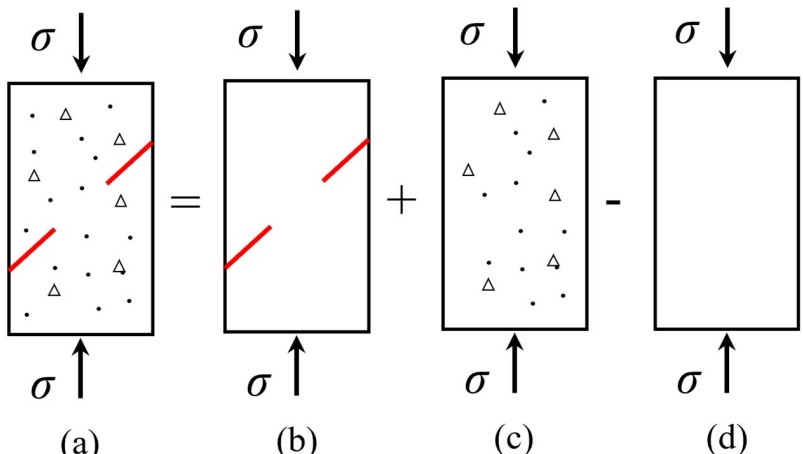

**Fig 8. Strain equivalent calculation: (a) Macro- and meso-defect composite rock mass; (b) Macro-defect rock mass; (c) Meso-defect rock; (d) rock without damage.**

The elastic moduli of the rocks in Fig 8(a)–8(d) are $\bar{E}_{12}$, $\bar{E}_1$, $\bar{E}_2$, and $\bar{E}_0$, respectively, and the strains generated under total stress are $\varepsilon_{12}$, $\varepsilon_1$, $\varepsilon_2$, and $\varepsilon_0$, respectively. Then,

$$\varepsilon_{12} = \varepsilon_1 + \varepsilon_2 - \varepsilon_0 \tag{2}$$

From the stress-strain relationship, Eq (2) is rewritten as

$$\frac{\sigma}{\bar{E}_{12}} = \frac{\sigma}{\bar{E}_1} + \frac{\sigma}{\bar{E}_2} - \frac{\sigma}{\bar{E}_0} \tag{3}$$

Let $D_1$ be the macro damage variable and $D_2$ the meso damage variable. The assumption of the strain equivalence principle proposed by Lemaitre shows

$$\left.\begin{array}{l} \bar{E}_{12} = \bar{E}_0(1 - D_{12}) \\ \bar{E}_1 = \bar{E}_0(1 - D_1) \\ \bar{E}_2 = \bar{E}_0(1 - D_2) \end{array}\right\} \tag{4}$$

Eq (4) is substituted into Eq (3) to obtain the variable expression of macro and mesoscopic composite damage of the cracked rock masses.

$$D_{12} = 1 - \frac{(1 - D_1)(1 - D_2)}{1 - D_1 D_2} \tag{5}$$

**3.2.2 Macroscopic damage variables of rocks.** Before the prefabricated cracks expand, additional strain energy $U_1$ caused by the prefabricated cracks is

$$U_1 = \int_0^A G\,dA = \frac{1}{E_0} \int_0^A \left(K_I^2 + K_{II}^2\right) dA \tag{6}$$

where $G$ is the energy release rate; $A$ the surface area of cracks; $K_I$ and $K_{II}$ are the first and second stress strength factors at the crack tip, respectively; elastic modulus of the cracked rocks $E_0 = E/(1 - v^2)$, where $E$ and $V$ are the elastic modulus and Poisson's ratio of intact rocks, respectively.

For the bilateral crack in Fig 1, $A = \pi Lc/2$, where $L$ is the diameter of the rock specimens and c is the crack length. Damage strain energy $W$ is expressed as follows under uniaxial stress [30]:

$$W = -\frac{\sigma^2}{2E(1 - D_1)^2} \tag{7}$$

$U^E$ is elastic strain energy per unit volume corresponding to stress, which can be expressed as

$$U^E = -(1 - D_1)W \tag{8}$$

Eq (7) is substituted into Eq (8) to obtain

$$U^E = \frac{\sigma^2}{2E(1 - D_1)} \tag{9}$$

When the rock masses do not contain any cracks, $D_1 = 0$. The increment of elastic strain energy per unit volume caused by the node is

$$\Delta U^E = U^E - U_0^E = \frac{\sigma^2}{2E}\left(\frac{1}{1 - D_1} - 1\right) \tag{10}$$

Assuming that $V$ is the volume of the rock masses, the increment of elastic strain energy caused by cracks should be equal to additional strain energy caused by prefabricated cracks.

$$\frac{1 - \nu^2}{E} \int_0^A \left(K_I^2 + K_{II}^2\right) dA = \frac{\sigma^2 V}{2E}\left[\frac{1}{1 - D_1} - 1\right] \tag{11}$$

From Eq (11), we obtain

$$D_1 = 1 - \frac{1}{1 + \frac{2}{V}\frac{(1 - \nu^2)}{\sigma^2} \int_0^A \left(K_I^2 + K_{II}^2\right) dA} \tag{12}$$

Normal stress and shear stress appear on the prefabricated crack interface under the uniaxial pressure. Normal stress closes the cracks, and shear stress causes rock masses to slide along the crack surface, denoted as

$$\sigma_\beta = \sigma \cos^2 \beta \tag{13}$$

$$\tau_\beta = \sigma \sin \beta \cos \beta \tag{14}$$

For closed cracks, the friction angle of the crack surface is set as $\varphi$. When the friction factor of the shear plane is $\mu = \tan \varphi$, tangential stress acting on the crack surface with inclination angle $\beta$ is expressed as

$$\tau_e = \tau_\beta - \mu\sigma_\beta, \ \tau_\beta \geqslant \mu\sigma_\beta \tag{15}$$

The wing cracks are due to local tensile stress at the crack tip caused by the frictional sliding of the crack surface. According to the research results of Lee and Ravichandran [31], stress strength factor $K_I$ and $K_{II}$ of wing cracks at the crack tip are modified as follows.

$$K_I = -\frac{2c\tau_e \sin \theta}{\sqrt{\pi(l + 0.27c)}} + \sigma_{\beta+\theta}\sqrt{\pi l} \tag{16}$$

$$K_{II} = -\frac{2c\tau_e \cos \theta}{\sqrt{\pi(l + 0.27c)}} - \tau_{\beta+\theta}\sqrt{\pi l} \tag{17}$$

where $l$ is the propagation length of the wing cracks; $\theta$ the propagation angle of the wing cracks at the crack tip (70.5°) [32]. When propagation length of wing cracks $l = 0$ (the critical condition), wing cracks propagate. The stress strength factor at the crack tip is substituted into Eq (12) to obtain the macroscopic damage variable of the rock masses due to the non-penetrating cracks on both sides.

The influence of the geometric characteristics of the non-penetrating cracks on the damage of the rock masses under the uniaxial impact load can be studied by changing inclination angle $\beta$ and penetration degree $k$ of non-penetrating cracks. Fig 9 shows the variation law of macroscopic damage value $D_1$ of the cracked rock masses with cracks' inclination angle $\beta$ at different penetration degrees ($k = 0.25, 0.35,$ and $0.45$).

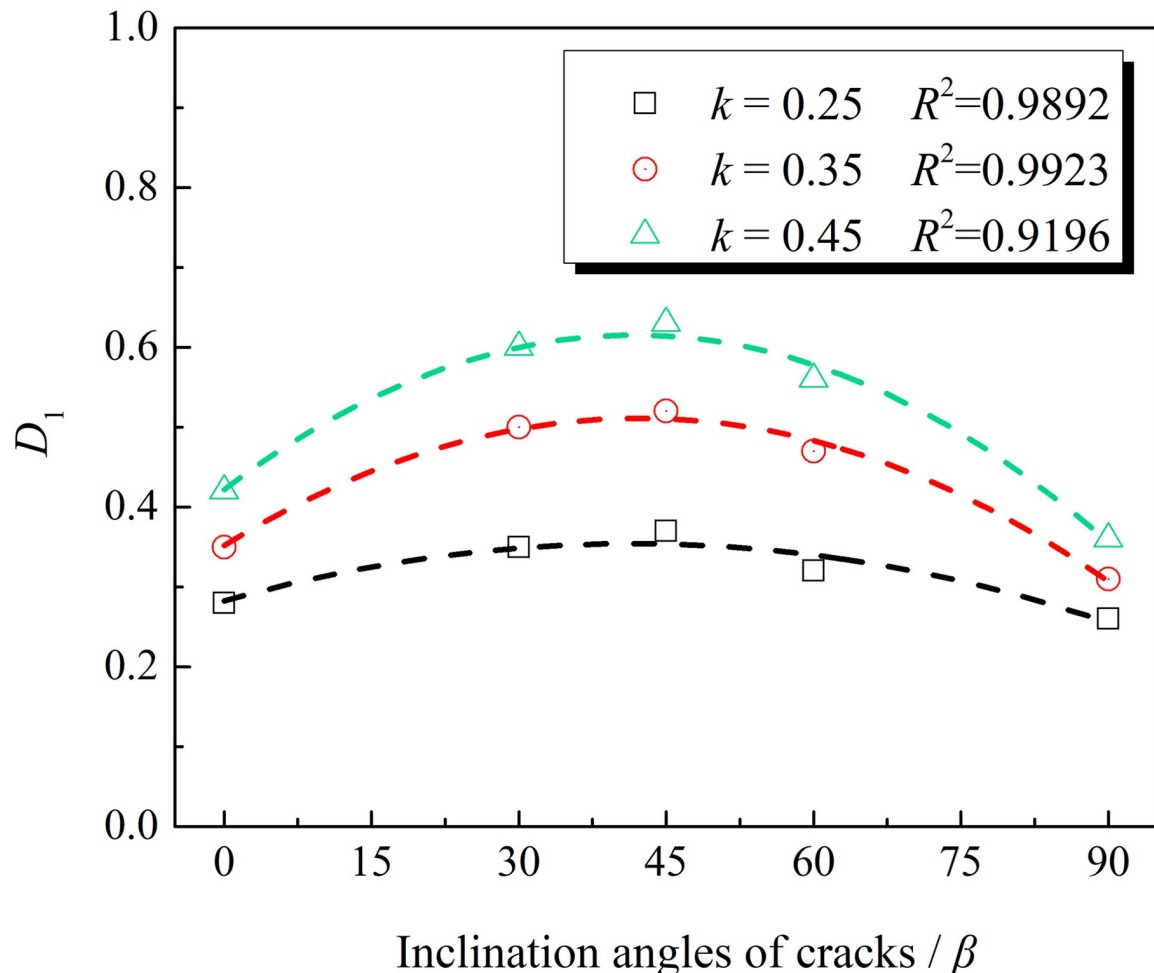

**Fig 9. Variation curves of the macroscopic damage variables of the cracked rock masses with cracks' inclination angles.**

In Fig 9, macroscopic damage value $D_1$ shows an arched distribution with the changed inclination angles of cracks. With the increased penetration of cracks, the arching height becomes more obvious. Damage is maximum when the cracks' inclination angle $\beta = 45°$ at the same crack penetration degree. Moreover, the difference in damage before and after 45° is small, and the overall change trend is parabolic. When the inclination angle of the crack is 0°, the prefabricated crack is tightly closed under axial stress; when the inclination angle is 90°, the prefabricated crack is consistent with the loading direction. Concentrated stress at the crack tip of the rock specimen is small under axial stress with relatively slow development, so damage is relatively light.

The research on penetrated-crack rock masses shows that [33] when the inclination angle $\beta$ is less than the friction angle on the crack surface, the cracked rock mass can be treated as a complete rock. Since the angle at which the crack surface is most likely to occur in the rock mass with penetrated cracks is $\alpha = 45° + \varphi_j/2$, shear slip failure along the crack surface mostly occurs. It is different from $\beta = 45°$, which is the most prone to failure of non-penetrating cracked rock masses in the work. Attention should be paid to the difference between penetration and non-penetrating cracks in studying the mechanical properties of cracked rock masses in engineering.

**3.2.3 Meso-damage variables of rocks.** Based on the principle of continuous factors and equivalent strain, the equation of damage variable $D_2$ satisfies:

$$D_2 = \frac{nS_0}{NS_0} = \frac{n}{N} \tag{18}$$

where $n$ is the number of microelements with damaged micro units; $N$ the total number of microelements in the rocks; $S_0$ the area of the microelements. When the rocks are subjected to external loads, their damage process is constantly evolving, and the micro-elements with initial damage continue to multiply.

From statistical damage mechanics, the various microscopic defects distributed in the rocks are random damage, and the probability density function is as follows [34].

$$P(\varepsilon) = \frac{m}{k} \left(\frac{\varepsilon}{k}\right)^{m-1} e^{-\left(\frac{\varepsilon}{k}\right)^m} \tag{19}$$

where $P(\varepsilon)$ is the rocks' micro-element strength distribution function; $\varepsilon$ the random-distribution variable of micro-element strength as well as the strain variable of the rocks due to the adopted strain strength theory; $m$ and $k$ are the shape parameter and scale of Weibull distribution.

When a certain strain level is loaded, the number of damaged elements is

$$n = \int_0^\varepsilon NP(\varepsilon)d\varepsilon = N\left[1 - e^{-\left(\frac{\varepsilon}{k}\right)^m}\right] \tag{20}$$

Eq (20) is used to obtain the mesoscale damage evolution equation of the loaded rocks with strain as the control variable of damage evolution, denoted as

$$D_2 = 1 - e^{-\left(\frac{\varepsilon}{k}\right)^m} \tag{21}$$

where $m$ and $k$ can be obtained by fitting the stress-strain relationship curve in the test.

## 3.3 Dynamic-damage constitutive model

The dynamic impact experiments of the rock masses with non-penetrating cracks show that the macro damage of the cracked rock masses softens the mechanical properties of the rock masses, and the strain rate hardens the mechanical properties. According to the definition of macro and mesoscopic composite damage variables of cracked rock mass in Section 3.2.1, macroscopic damage and mesoscopic damage are connected in parallel to form macro and mesoscopic composite damage body $D_{12}$ according to the principle of strain equivalence. Then, the Maxwell body is connected in parallel to construct a dynamic damage model of the rock masses with non-penetrating cracks (see Fig 10).

The Maxwell body has instantaneous deformations, constant rate creep, and relaxation, and the constitutive equation is expressed as

$$\dot{\varepsilon} = \frac{\dot{\sigma}_M}{E_M} + \frac{\sigma_M}{\eta_0} \tag{22}$$

where $\sigma_M$ is Maxwell stress; $E_M$ the Maxwell elasticity modulus; $\dot{\varepsilon}$ the strain rate; $\dot{\sigma}_M$ the loading rate; $\eta$ the experimental parameter.

The Laplace transforms of Eq (22) can remove loading rate $\dot{\sigma}_M$.

$$L(\dot{\varepsilon}(t)) = L\left(\frac{\dot{\sigma}_M}{E_M}(t)\right) + L\left(\frac{\sigma_M}{\eta}(t)\right) \tag{23}$$

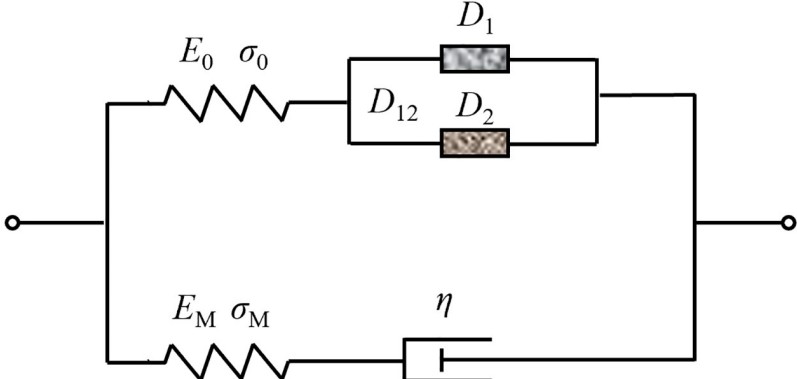

**Fig 10. Dynamic model of viscoelastic composite damage.**

Combined with boundary conditions, $\varepsilon(0) = 0$, $\sigma_M(0) = 0$. Eq (22) is transformed into

$$\sigma_M(s) = \frac{E_M \dot{\varepsilon}(t)}{s(s + E_M/\eta)} \tag{24}$$

Eq (24) is subjected to the inverse Laplace transforms to obtain

$$\sigma_M(t) = \dot{\varepsilon}(t)\eta \left[1 - \exp\left(-\frac{E_M \varepsilon(t)}{\eta \dot{\varepsilon}(t)}\right)\right] \tag{25}$$

According to the definition of macro and mesoscopic composite damage variables of the cracked rock masses as well as Hooke's law, the rock-damage constitutive equation is expressed as

$$\varepsilon_D = \frac{\sigma_D}{E_0(1 - D_{12})} \tag{26}$$

where $\sigma_D$ is the damaging stress; $E_0$ is the same as above.

From the series-parallel relationship of each element in the model in Fig 10,

$$\sigma = \sigma_D + \sigma_M \tag{27}$$

$$\varepsilon = \varepsilon_D = \varepsilon_M \tag{28}$$

Eqs (25)–(28) are jointly solved to obtain the dynamic-damage constitutive equation of cracked rock masses.

$$\sigma = E_0(1 - D_{12})\varepsilon(t) + \dot{\varepsilon}(t)\eta \left[1 - \exp\left(-\frac{E_M \varepsilon(t)}{\eta \dot{\varepsilon}(t)}\right)\right] \tag{29}$$

Eqs (5), (12) and (21) are substituted into Eq (29) to obtain the dynamic damage model of the cracked rock masses.

## 4 Constitutive model verification and parameter analysis

### 4.1 Test curve fitting and verification

Since there are many constitutive-equation fitting parameters in Eq (29), trials are necessary with the experimental test data before fitting to determine the corresponding fitting and test

**Table 1. Parameters of the granite used in this test.**

| $\rho$(kg/m³) | $E$(GPa) | $v$ | $k$ | $m$ | $\eta$ | $E_M$(GPa) |
|---|---|---|---|---|---|---|
| 2670 | 28.64 | 0.28 | 0.02 | 1.4 | 0.026 | 6.5 |

parameters. Finally, the theoretical stress-strain curve is obtained according to the fitting parameters (see Table 1 for the complete granite test data and fitting parameters).

The dynamic damage model of the rock masses shows that $k$, $m$, $\eta$, and $E_M$ in the model are determined by fitting the experimental data, and each parameter significantly affects the dynamic constitutive relationship of the rock masses. When other parameters are fixed values, $k$, $m$, $\eta$, and $E_M$ are changed to obtain the stress-strain relationship corresponding to the macro and mesoscopic composite damage body of the cracked rock masses (see Fig 11).

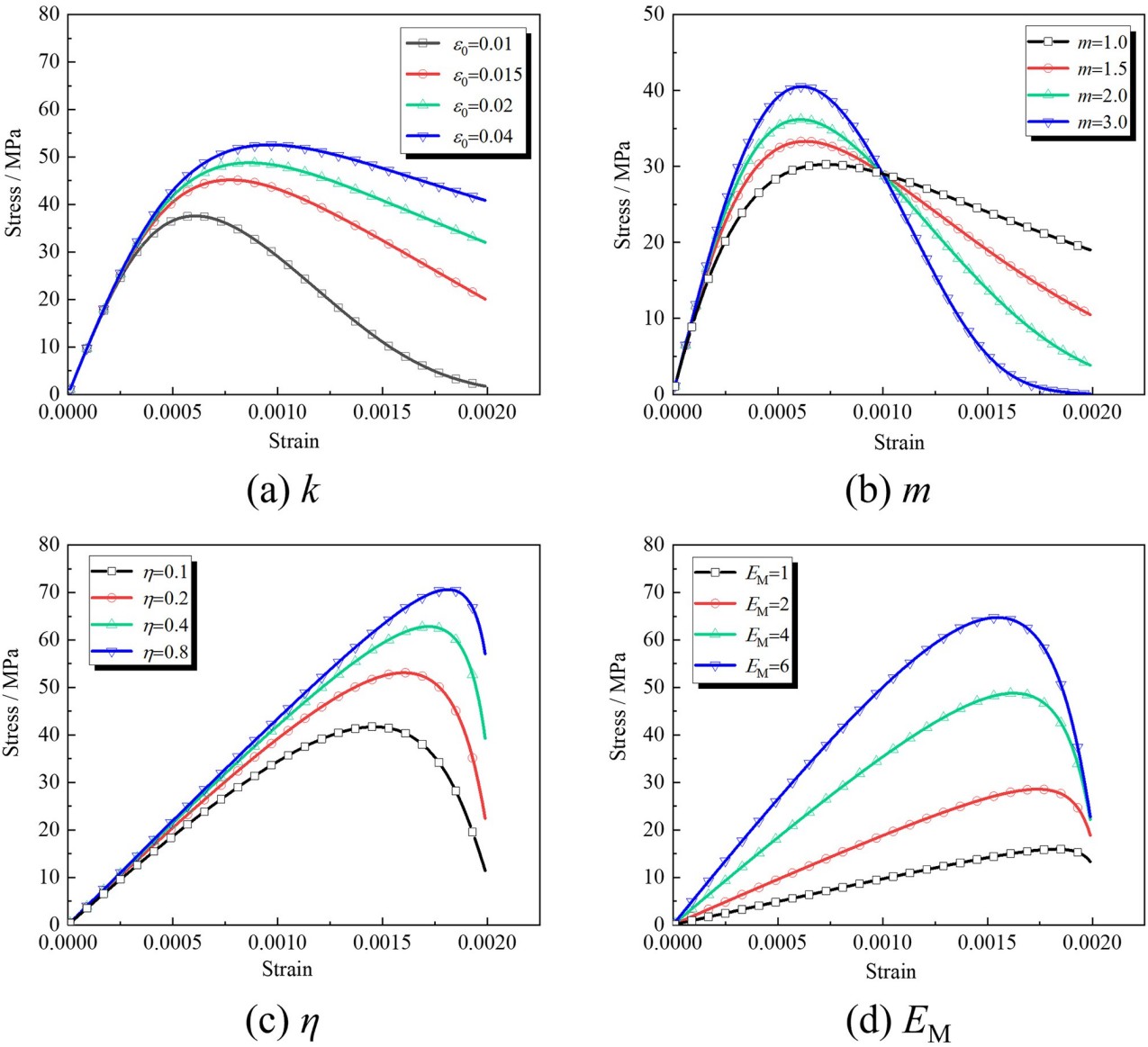

**Fig 11. Influence of fitting parameters on constitutive relation.**

Fig 11(a) shows that the influence of *k* on the constitutive relationship is mainly manifested at the development stage of the inelastic micro-element cracks, and the peak strength and corresponding strain of rocks increase with increased *k*. Parameter *m* in Fig 11(b) significantly affects the peak strength of rocks and the curvatures at the pre-and post-peak inelastic deformation stages. The curvatures increase correspondingly with increased *m*, indicating that *m* reflects the strength of microelements in the rocks. Fig 11(c) shows that the peak strength of the rocks and the curvature at the inelastic deformation stage increase with increased *η*, and the dynamic viscous effect of the rocks increases accordingly. The larger *η* is, the larger the growth rate of the viscous effect, indicating that *η* reflects the correlation degree of the rocks' strain rate effect. $E_M$ is vital for the viscous effect of Maxwell bodies. Different from the effect of *η*, increased rocks' peak strength increases with increased $E_M$. The curvatures of the constitutive relation curve at the pre-peak and post-peak stages are correspondingly larger, indicating that $E_M$ also reflects the concentration ratio of the viscous effect. It presents the correlation and concentration of the rocks' strain rate effect.

According to the established constitutive equation of the cracked rocks under repeated impact loading, the physical and mechanical parameters of the cracked rock masses are substituted into Eq (29) to obtain the theoretical model curve of the cracked rock masses. Fig 12 compares the stress-strain curves of the rock masses with 30° non-penetrating cracks obtained by the theoretical model under the uniaxial dynamic load and the experimental data. Table 2 shows the variation of fitting parameters with the number of impacts (*n*).

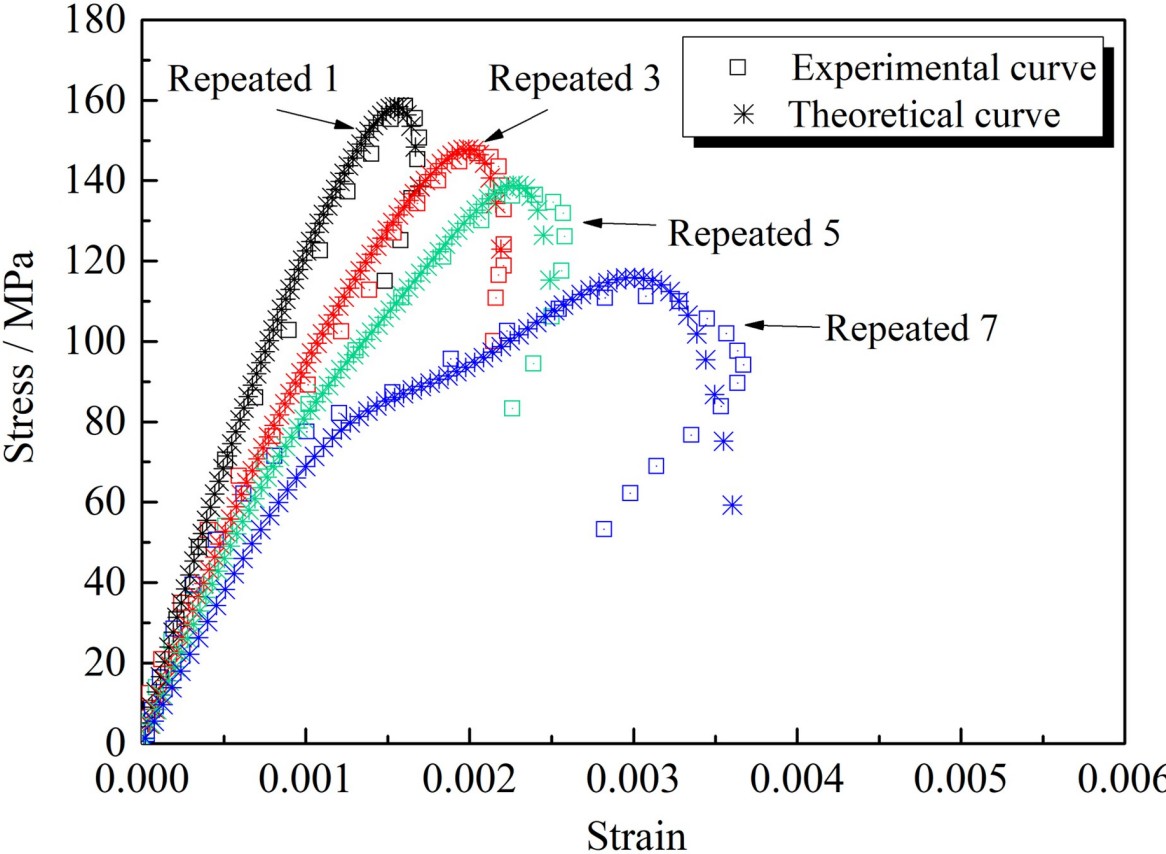

**Fig 12. Comparison between experimental results and theoretical calculation results.**

**Table 2. The variation of fitting parameters with the number of impacts.**

| Fitting parameters | Repeated 1 | Repeated 3 | Repeated 5 | Repeated 7 | Fitting formula | Determination coefficient / $R^2$ |
|---|---|---|---|---|---|---|
| $k$ | 0.012 | 0.01 | 0.008 | 0.0075 | $k = 0.0135-0.00153n+9.37\times10^{-5}n^2$ | 0.9734 |
| $m$ | 1.2 | 1.4 | 2 | 2.8 | $m = 1.1825-0.03n+0.0375n^2$ | 0.9961 |
| $\eta$ | 0.028 | 0.025 | 0.025 | 0.008 | $\eta = 0.01789+0.5\times10^{-4}n-1.875\times10^{-4}n^2$ | 0.9535 |
| $E_M$(GPa) | 9.5 | 10 | 10.1 | 10.5 | $E_M = 10.85-0.3n$ | 0.9595 |

In this test, multiple impacts at the same impact speed were carried out on the cracked rock specimens. The impact number of rock specimens under different crack forms are different, which changes the responses of the fitting parameters in the constitutive equation. The experimental results are consistent with the model's calculation results. Determination coefficient $R^2$ is above 0.95, indicating the good fitting effect of the model.

The definition of macro and mesoscopic composite damage variables of cracked rocks is used to obtain the damage evolution curves of the rock specimens ($k = 0.25$) with a cracks' inclination angle of 30° and the intact rock specimens ($k = 0$) (see Fig 13). As strain increases, the load damage evolution curves of the cracked and intact specimens coincide. The analysis

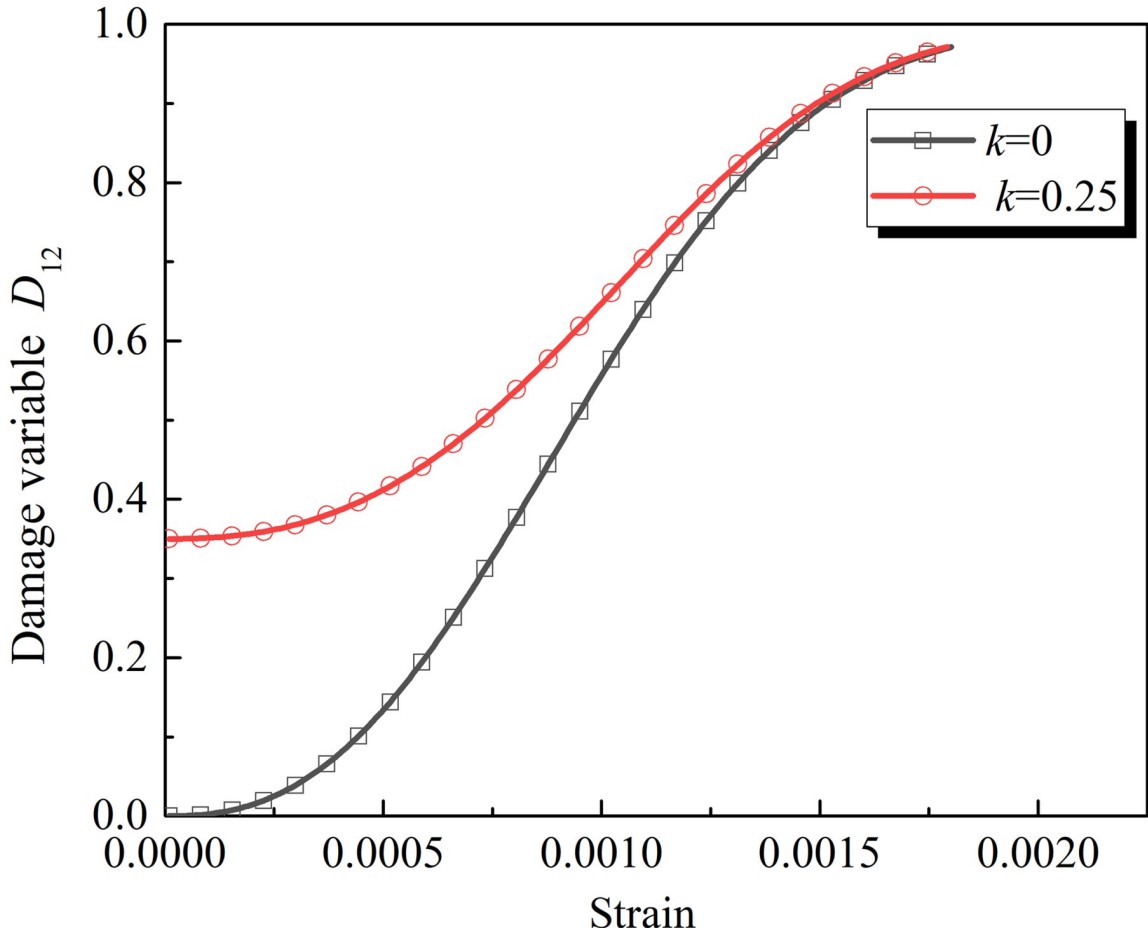

**Fig 13. Damage variable and strain relationship curves.**

shows that the intact rocks produce macroscopic cracks under the compressive loading, that is, the intact rocks are transformed into ones containing macroscopic and mesoscopic defects. Its mechanical behaviors are also affected by macro and mesoscopic defects. Therefore, the mechanical behaviors of the intact rocks are similar to those of prefabricated-crack rocks after cracks appear.

## 4.2 Influence of cracks' inclination angles

The parameters of different cracks' inclination angles are substituted into the constitutive equation of cracked rocks under repeated impact loading to obtain the theoretical model curves of different cracked rock masses. Fig 14 compares the stress-strain curves of the rock masses with non-penetrating cracks obtained by the theoretical model and the experimental

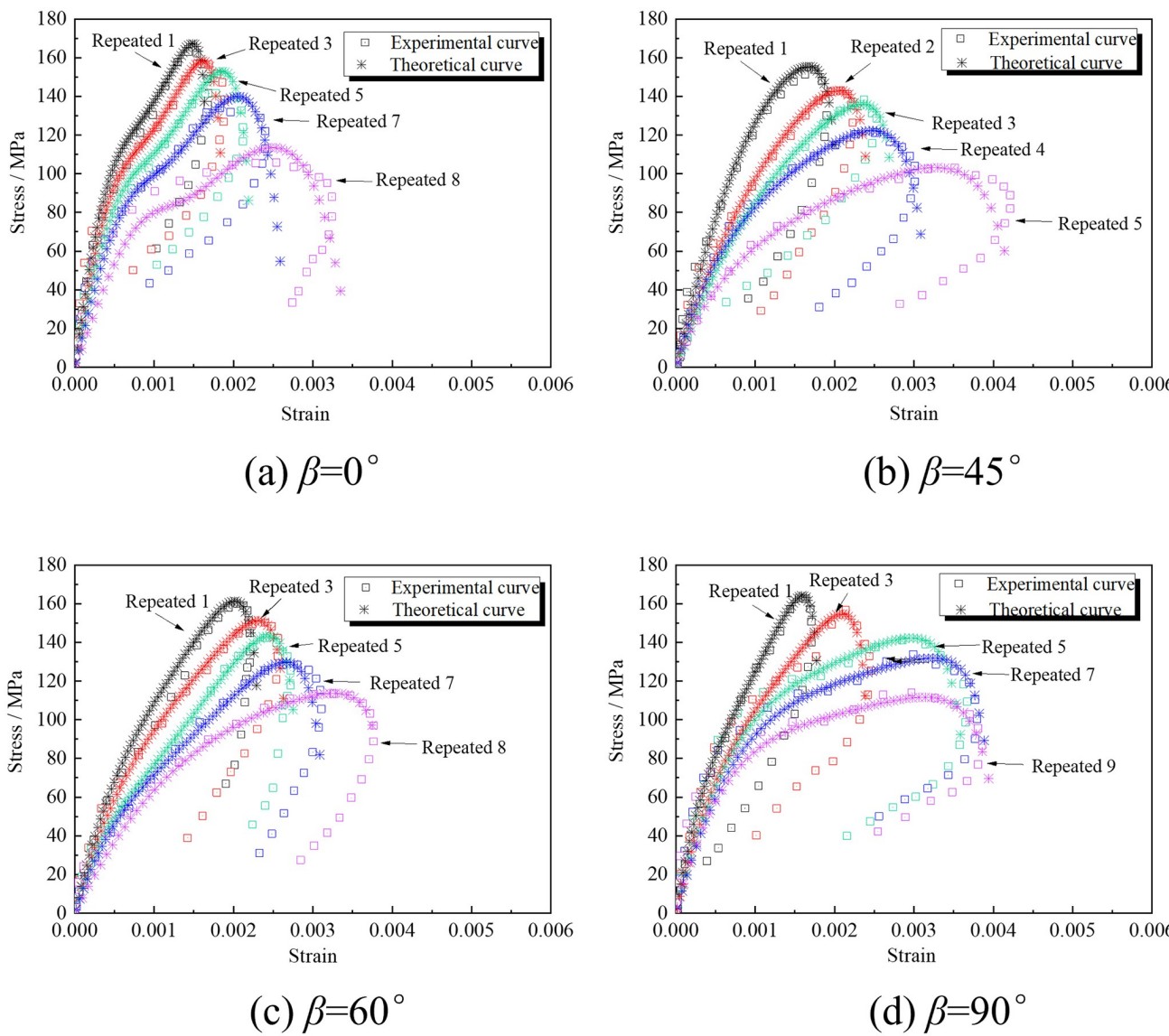

**Fig 14. Comparison between experimental results and theoretical calculation results of different cracked rock masses.**

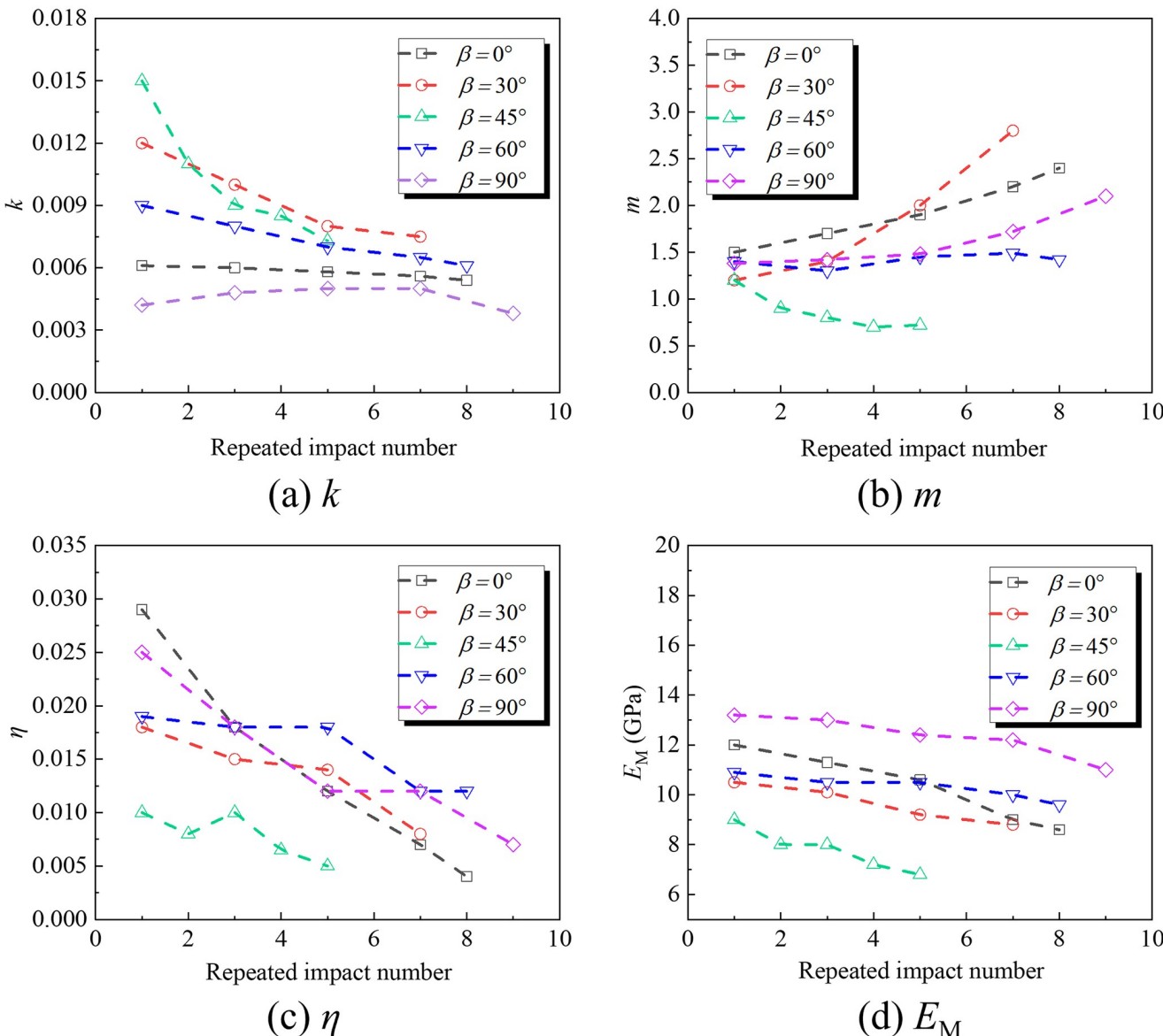

**Fig 15. Variation of fitting parameters of different cracked rock masses with impact number.**

data. Fig 15 presents the variation of fitting parameters with the number of impacts. The two are in good agreement, indicating that the damage model based on the coupling of macro- and meso-defects proposed in the work can better reflect the strength and deformation of the rock masses with non-penetrating cracks.

Fig 15 shows that $k$, $\eta$, and $E_M$ decrease with the increased impacts. In the process of multiple impacts, the inelastic deformation of the rocks is more significant than the elastic deformation, and their peak strength decreases as the number of repeated impacts increases. Besides, the dynamic viscous effect of the rocks is reduced and the concentration degree increases. $m$ increases with the increased impacts, indicating that a certain number of impacts can improve the concentration degree of micro-element strength in the rocks. However, the cracked rocks of 45° are damaged under fewer impact numbers, which fails to reflect the trend.

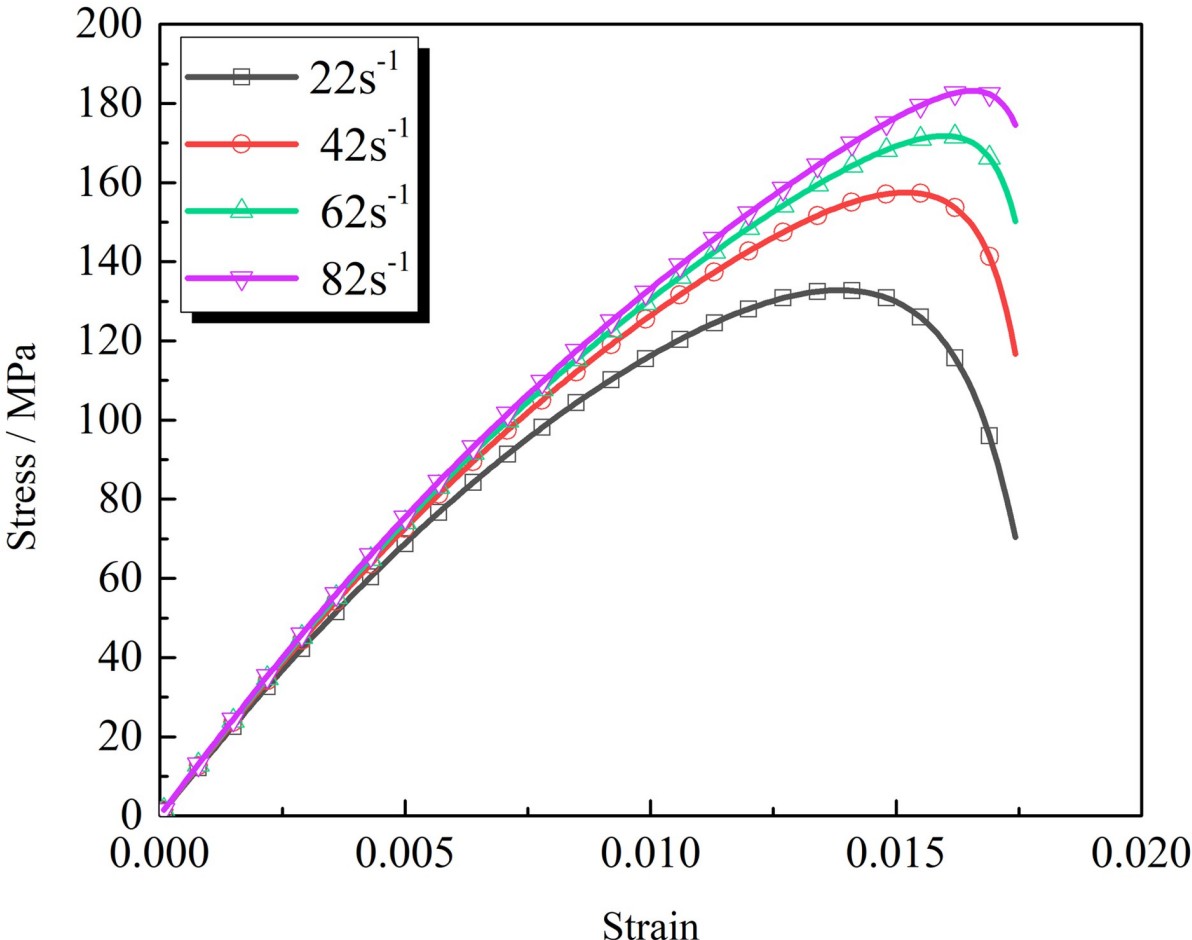

**Fig 16. Effect of strain rate on stress-strain relationship of rock masses.**

With the increased inclination angles of cracks, $\eta$ and $E_M$ first decrease and then increase. The 0° and 90° cracked rocks have a larger viscous effect and concentration degree, so their ability to resist impacts is stronger and they have stronger brittleness when failing. $k$ first increases and then decreases with the increased inclination angle. The 45° cracked rocks are the most significant in the crack development of the inelastic micro-body, and the 90° cracked rocks have the most stable crack development. Since inclination angle $\beta$ in the calculation model reaches 90°, normal stress and tangential stress on the crack surface are both zero, resulting in no obvious difference between the mechanical properties of the rock masses with 90° fractures and the intact rock.

From the stress-strain curves obtained by fitting, cracks with different inclination angles have a significant effect at the yield stage instead of at the elastic deformation stage of the rock masses. The stress strength factor at the crack tip has not reached the crack toughness of rocks at the elastic deformation stage, and the damage to the rock masses caused by cracks is mainly the closing deformation of the crack surface. When the factor at the crack tip reaches the fracture toughness, the cracks propagate. The macro and mesoscopic coupling damage of rock masses increases significantly, so the stress-strain curve slows down until failure.

### 4.3 Effect of the strain rate

In the test, the repeated impact air pressure was 0.18 MPa, and the load strain rate of the specimen under the first impact load was approximately 42 s$^{-1}$. The impact number and the mechanical properties of the rock specimens are different with the changed impact loads on the rocks. Taking the rocks with a cracks' inclination angle of 30˚ as an example, the load-strain rates were set to 22, 42, 62, and 82 s$^{-1}$, respectively, to obtain the rock stress-strain relationship (see Fig 16).

The strain rate effect of the rock masses under different load strain rates was small at the elastic deformation stage but significant at the crack propagation stage. The peak strength of the rock mass specimens increased gradually with the increased load-strain rate. The peak strength under four different strain-rate loads were 132, 157, 172, and 183 MPa, respectively, and corresponding peak strain and total strain increase. It is consistent with the deformation law of the rock masses under different load-strain rates in References [35,36].

## 5 Conclusion

1. Crack mechanics and energy theory were used to derive a calculation method for macroscopic damage variables, which considered the geometric parameters and mechanical parameters of cracks. The macroscopic damage changes of the rocks obtained by calculation were consistent with that of the cracked rocks in the impact test. The damage of the specimens with a cracks' inclination angle of 45˚ was the most serious; that with an inclination angle of 90˚ was the minimum, and its impact resistance was closer to the intact rocks.

2. A dynamic impact-damage constitutive model considering the macro and mesoscopic defects of cracked rock masses was established using model deformation. The model reflected the whole process of the deformation of the cracked rock masses under the impact dynamic loads, indicating that the model in the work could describe the impact mechanical properties of the rock masses with non-penetrating cracks.

3. The effects of the crack geometry and strain rate on the damage and stress-strain behaviors of cracked rock masses were discussed using the model, finding that the existing cracks weakened the strength and stiffness of the rock masses. The mechanical behaviors of intact rocks were close to those of cracked rocks, with the difference decreasing after peak strength. They even tended to be consistent.

4. The fitting parameters in this constitutive model reflected the influence of crack structures on the mechanical behaviors of rock masses. With the increased cycles, it showed a strong regularity. The impact test results were used for fitting and determining the parameters in the model.

## Supporting information

**S1 Dataset.**
(DOCX)

## Author Contributions

**Data curation:** Jie Zhang.

**Formal analysis:** Jie Zhang, Xu Wu.

**Investigation:** Jie Zhang, Xu Wu.

**Methodology:** Jie Zhang, Xu Wu.

**Supervision:** Xu Wu.

**Writing – original draft:** Jie Zhang.

**Writing – review & editing:** Xu Wu.

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
