## [Decision Letter · Decision Letter 0]

13 Jun 2023

PONE-D-23-11643Damage evolution and constitutive model of the rock masses with non-penetrating cracks under cyclic impact disturbancesPLOS ONE

Dear Dr. Zhang,

Thank you for submitting your manuscript to PLOS ONE. After careful consideration, we feel that it has merit but does not fully meet PLOS ONE’s publication criteria as it currently stands. Therefore, we invite you to submit a revised version of the manuscript that addresses the points raised during the review process.

The paper requires major changes (see reviewers opinion below)

We look forward to receiving your revised manuscript.

Kind regards,

Pawel Klosowski, D.Sc.

Academic Editor

PLOS ONE

Journal Requirements:

"NO"

"NO authors have competing interests"

Reviewers' comments:

Reviewer's Responses to Questions

**Comments to the Author**

1. Is the manuscript technically sound, and do the data support the conclusions?

Reviewer #1: Partly

2. Has the statistical analysis been performed appropriately and rigorously? 

Reviewer #1: I Don't Know

3. Have the authors made all data underlying the findings in their manuscript fully available?

Reviewer #1: Yes

4. Is the manuscript presented in an intelligible fashion and written in standard English?

Reviewer #1: No

5. Review Comments to the Author

Reviewer #1: The paper presents experimental and theoretical considerations on the influence of macro and mesoscopic initial defects on the mechanical properties of granite, as well as on the evolution of damage in cracked granite subjected to repeated loads with high strain rates.The main objective of the presented researches was to develop the dynamic-damage constitutive model for cracked granite, and determination of material parameters depending on the number of impacts.

On the basis of the obtained results, the Authors prove the validity of the developed constitutive model, which well describes the deformation process of cracked granite. In my opinion, it is the most valuable and of interest to the scientific community.

The manuscript is generally well written and organized. However, there are some shortcomings regarding the presented work completeness, which in my opinion should be addressed before the paper can be considered for publication.

1. I recommend that the title of the manuscript be slightly changed. Instead of the term "cyclic-impact disturbances", which may be associated with fatigue tests, I suggest replacing it with the phrase "repeated impact loading". Generally speaking, the term "disturbances" does not fit the issue of the manuscript. I recommend replacing it with "loading" throughout the paper.

2. There is no information in the manuscript about the number of material specimens made. In other words, how many specimens were used for quasi-static and high strain rate tests and how many test repetitions were made under given experimental conditions (section 2.1).

3. Fig. 1a - no axis of symmetry and no symbol indicating the diameter of a cylindrical specimen. In addition, prefabricated crack surface S2 was marked incorrectly in Fig. 1b.

4. Incorrect terminology: line 108 and 110, Fig. 2; "warhead", "spindle punch"; the commonly used term is "striker bar" or more accurately "cone-shaped striker". I also propose to replace the term "impact times" with "impact number".

5. Line 103; "....SHPB device with a diameter of 50 mm...." diameter of what?

6. Line 117; "...with certain strength." Are you sure it's about "strength" and not "force"?

7. Line 118; force in MPa?

8. Please explain how the material specimen was loaded with an axial static force on the SHPB stand.

9. Fig. 3 has already been presented in [27]. I believe that a literature reference should be added in the caption of the Figure.

10. Fig. 4; Incorrectly marked wave profiles. The descriptions of the reflected and transmitted waves should be interchanged.

11. line 162; I recommend adding a description of determining the dynamic deformation modulus, especially since in work [28] I could not find a description of determining the above-mentioned parameter.

12. Both quasi-static and dynamic strength tests were carried out using specimens with a length to diameter ratio of L/d = 2. According to the recommendations presented in the literature, dynamic tests of rocks, concrete and similar materials under the SHPB test conditions should be performed on a specimens with an L/d ratio of 1. This ratio is important because of the need to achieve an equilibrium stress state in the specimen during dynamic loading. This is the basic methodological requirement of the SHPB technique. Therefore, in order to recognize that the results presented in the manuscript are credible, the manuscript should be supplemented with an analysis of the state of equilibrium stress in the applied loading conditions. This is my most important remark, which should be taken into account in order to recognize that the developed constitutive model and the determined material parameters are correct.

13. The sentence (lines 166 and 167) is incomprehensible.

14. The sentence (lines 184 and 185) is rather redundant as it does not refer to the content of the paragraph.

15. Line 430; statement "...their peak strength decreases with increased strain." is not true.

16. Line 455; How was the strain rate value determined? From Fig. 4 it can be seen that the strain rate during the SHPB experiments was not constant. So what was the strain rate estimation procedure?

17. Line 465 and 466; I believe that giving the peak strength value with an accuracy of two decimal places is a mistake. The SHPB technique does not allow for such exact determination of the stress in the specimen material.

18. Line 490 and 491; statement "Higher coefficient of determination R2 indicates a better fitting effect" is obvious and redundant in the chapter where the conclusions are presented.

19. I am not an expert in English, but grammatical and editorial errors and incorrectly used phrases. etc., are abundant in the manuscript.

6. PLOS authors have the option to publish the peer review history of their article (what does this mean?). If published, this will include your full peer review and any attached files.

Reviewer #1: No

---

## [Author Response · Author response to Decision Letter 0]

29 Jun 2023

The revisions have been made according to the reviewer's requirements.

---

## [Decision Letter · Decision Letter 1]

10 Jul 2023

Damage evolution and constitutive model of the rock masses with non-penetrating cracks under repeated impact loading

PONE-D-23-11643R1

Dear Dr. Zhang,

We’re pleased to inform you that your manuscript has been judged scientifically suitable for publication and will be formally accepted for publication once it meets all outstanding technical requirements.

Kind regards,

Pawel Klosowski, D.Sc.

Academic Editor

PLOS ONE

Additional Editor Comments (optional):

Reviewers' comments:

Reviewer's Responses to Questions

**Comments to the Author**

1. If the authors have adequately addressed your comments raised in a previous round of review and you feel that this manuscript is now acceptable for publication, you may indicate that here to bypass the “Comments to the Author” section, enter your conflict of interest statement in the “Confidential to Editor” section, and submit your "Accept" recommendation.

Reviewer #1: All comments have been addressed

2. Is the manuscript technically sound, and do the data support the conclusions?

Reviewer #1: Yes

3. Has the statistical analysis been performed appropriately and rigorously? 

Reviewer #1: Yes

4. Have the authors made all data underlying the findings in their manuscript fully available?

Reviewer #1: Yes

5. Is the manuscript presented in an intelligible fashion and written in standard English?

Reviewer #1: Yes

6. Review Comments to the Author

Reviewer #1: All comments have been taken into account. In particular, the authors supplemented the manuscript with Fig. 5, which proves that the conditions of the SHPB experiment were selected in such a way that the basic methodological requirement of the SHPB technique was met.

7. PLOS authors have the option to publish the peer review history of their article (what does this mean?). If published, this will include your full peer review and any attached files.

Reviewer #1: No

---

## [Editor Report · Acceptance letter]

7 Sep 2023

PONE-D-23-11643R1 

Damage evolution and constitutive model of the rock masses with non-penetrating cracks under repeated impact loading 

Dear Dr. Zhang:

I'm pleased to inform you that your manuscript has been deemed suitable for publication in PLOS ONE. Congratulations! Your manuscript is now with our production department. 

Kind regards, 

on behalf of

Prof. Pawel Klosowski 

Academic Editor

PLOS ONE